# A Targeted Bioinformatics Assessment of Adrenocortical Carcinoma Reveals Prognostic Implications of GABA System Gene Expression

**DOI:** 10.3390/ijms21228485

**Published:** 2020-11-11

**Authors:** Erika L. Knott, Nancy J. Leidenheimer

**Affiliations:** Department of Biochemistry and Molecular Biology, Louisiana State University Health Shreveport, Shreveport, LA 71103, USA; eknot1@lsuhsc.edu

**Keywords:** adrenocortical carcinoma, GABA shunt, *ABAT*, GABA_A_ receptors, *GABRD*, metabolic heterogeneity, NCI-H295R, cBioPortal, TCGA, cancer

## Abstract

Adrenocortical carcinoma (ACC) is a rare but deadly cancer for which few treatments exist. Here, we have undertaken a targeted bioinformatics study of The Cancer Genome Atlas (TCGA) ACC dataset focusing on the 30 genes encoding the γ-aminobutyric acid (GABA) system—an under-studied, evolutionarily-conserved system that is an emerging potential player in cancer progression. Our analysis identified a subset of ACC patients whose tumors expressed a distinct GABA system transcriptome. Transcript levels of *ABAT* (encoding a key GABA shunt enzyme), were upregulated in over 40% of tumors, and this correlated with several favorable clinical outcomes including patient survival; while enrichment and ontology analysis implicated two cancer-related biological pathways involved in metastasis and immune response. The phenotype associated with *ABAT* upregulation revealed a potential metabolic heterogeneity among ACC tumors associated with enhanced mitochondrial metabolism. Furthermore, many GABA_A_ receptor subunit-encoding transcripts were expressed, including two (*GABRB2* and *GABRD)* prognostic for patient survival. Transcripts encoding GABA_B_ receptor subunits and GABA transporters were also ubiquitously expressed. The GABA system transcriptome of ACC tumors is largely mirrored in the ACC NCI-H295R cell line, suggesting that this cell line may be appropriate for future functional studies investigating the role of the GABA system in ACC cell growth phenotypes and metabolism.

## 1. Introduction

Adrenocortical carcinoma (ACC) is a rare type of adrenal malignancy diagnosed in approximately 1 out of every 1 million people in the United States. The median 5-year survival rates for patients remain at 74%, 56%, and 37% for patients with localized ACC, regional metastases, and distant metastases, respectively [1]. Surgery is the primary treatment for ACC. Despite improvements in early and incidental detection [2], drug treatment options remain very limited [3]. 

ACC originates from neuroendocrine foci in the adrenal cortex [4]. The primary function of the adrenal cortex is to produce steroid hormones, including cortisol [5]. Excess serum cortisol is present in roughly a third of ACC patients [6], with about a quarter of patients showing symptomatic hypercortisolism [7]. Investigations into developing novel ACC treatments have largely targeted steroidogenic pathways. In preclinical models, inhibition of steroidogenesis enzymes [8,9] or manipulation of steroidogenic factor-1 expression levels [10] affects cell proliferation and invasion, as well as tumor volume. However, inhibition of steroidogenesis enzymes with clinically available drugs such as ketoconazole, metyrapone, and aminoglutethimide does not affect the progression of the ACC tumors; thus these drugs are recommended only as adjuvant treatment and/or for symptom management [6]. Radiotherapy targeting steroidogenesis enzymes results in moderate success in a very small ACC patient cohort [11]. While treatment with the adrenolytic agent mitotane results in ACC tumor regression in roughly half of patients across multiple studies, this drug is severely toxic [6]. 

It has been suggested that bioinformatics approaches may be fruitful for identifying novel targets for ACC therapeutics [12]. Unsurprisingly, unsupervised bioinformatics studies show that genes that are differentially expressed between ACC tumors and normal/noncancerous adrenal tissues are largely those encoding proteins involved in cell cycle regulation or cell division [13,14,15,16,17]. The identification of genes that are differentially expressed among ACC patients with distinct clinical outcomes may prove a more advantageous approach to identifying novel targets. Indeed, prognostic ACC genes have been identified by clustering patients by disease-free survival [18]. Here, we use this approach to examine gamma-aminobutyric acid (GABA) system genes in ACC patient tumors. 

GABA, a non-proteinaceous amino acid, is present in normal adrenal gland [19], the cortex of which expresses multiple GABA system transcripts and proteins [20]. Although primarily studied in brain for its role in inhibitory neurotransmission, GABA is thought to modulate steroid production in the adrenal cortex [21]. Here we examine the expression of 30 genes across 14 chromosomes that encode the enzymes, receptors, and transporters of the GABA system. GABA is predominately synthesized from glutamate in a decarboxylation reaction catalyzed by glutamic acid decarboxylase (GAD67 or GAD65, encoded by *GAD1* and *GAD2* respectively). Subsequently, GABA undergoes transamination catalyzed by γ-aminobutyrate aminotransferase (protein and transcript ABAT; aka GABA-transaminase/ GABA-T) to produce succinic semialdehyde (SSA) (with the simultaneous conversion of α-ketoglutarate to glutamate). SSA is then dehydrogenated into the tricarboxylic acid (TCA) cycle intermediate succinate in a reaction catalyzed by succinic semialdehyde dehydrogenase (SSADH, encoded by *ALDH5A1*). This pathway, known as the GABA shunt, constitutes a bypass for two steps of the TCA cycle. GABA also activates two types of receptors, ionotropic heteropentameric GABA_A_ receptors and metabotropic GABA_B_ receptors. Additionally, five GABA transporters carry GABA across plasma membranes or package GABA into vesicles. Although GABA is present in ACC tumors [19], the role of the GABA system in this cancer has not yet been investigated. Importantly, drugs that target many components of the GABA system are in clinical use to treat neurological and psychiatric disorders. If the GABA system is discovered to be important for ACC progression, such drugs could potentially be repurposed for treating this cancer. 

Here, bioinformatics was used to assess genomic alterations, transcript expression, and methylation status of the 30 GABA system genes in ACC tumors within The Cancer Genome Atlas (TCGA) dataset. The association of these parameters with patient prognosis and clinical attributes was evaluated, and enrichment analysis was used to examine genes and biological pathways associated with the GABA system phenotype. This examination identified that ACC tumors express a robust GABA system transcriptome. In particular, upregulation of tumoral *ABAT* transcripts is correlated both with favorable clinical outcomes and the overexpression of two biological pathways relevant to limiting cancer progression. Transcripts encoding GABA_A_ and GABA_B_ receptor subunits, as well as GABA transporters, are also present in ACC tumors, with differential prognostic values. Lastly, using the National Center for Biotechnology Information’s Gene Expression Omnibus, as well as performing real-time polymerase chain reaction (RT-PCR) experiments, we determined that the GABA system transcriptome in the ACC cell line NCI-H295R largely mirrors that of ACC tumors. Ultimately, this work supports future functional investigations into the role of the GABA system in ACC. 

## 2. Results

We accessed data from the ACC TCGA dataset using the cBioPortal user interface. ACC patient demographic information is in Appendix A. For each of the 30 GABA system genes (Table 1), ACC tumors were examined for mutations, copy number alterations (CNA), RNA-seq expression levels, and methylation status. Although infrequent mutations and CNA were observed in some of these genes, none had mutations, homozygous deletions, or high-level amplifications that exceeded 4%. We therefore primarily focused on gene transcript expression unless otherwise indicated. To this end, violin plots showing expression levels of multiple GABA system protein-encoding transcripts display a solid line at 3 log_2_ RNA-Seq by Estimation Maximization (RSEM) with shading below, indicating our cutoff for transcript expression (see Methods for rationale).

### 2.1. The GABA Shunt

ACC tumors expressed *GAD1*, *ABAT*, and *ALDH5A1* transcripts at 8.21, 10.53, and 8.80 log_2_ RSEM median values, respectively, but expressed minimal *GAD2* transcripts (−0.16 log_2_ RSEM) (Figure 1a). *ALDH5A1* was highly expressed with little inter-patient variability, while both *GAD1* and *ABAT* had a wide range of expression. A subset of tumors displayed upregulated transcripts (*z*-score ≥ 1) for *GAD1* (9 out of 78) and *ABAT* (33 out of 78) at 12% and 42% of tumors, respectively. As shown in Figure 1b, transcript levels for these two genes were significantly higher in tumors displaying upregulation vs unaltered expression (*GAD1*: avg ± SEM = 10.9 ± 0.2 vs 7.6 ± 0.2 log_2_ RSEM; *ABAT*: avg ± SEM = 12.9 ± 0.1 vs 8.8 ± 0.3 log_2_ RSEM). Transcript expression of these genes was positively correlated (Spearman *r* = 0.44; *q* = 5.8 × 10^−4^). Neither *GAD1* nor *ABAT* displayed downregulation (*z*-score ≤ −1). 

Due to the striking percentage of patient tumors with upregulated *ABAT*, we evaluated its association with patient outcome. Kaplan-Meier estimate analysis revealed a significantly higher overall survival estimate in patients with upregulated tumoral *ABAT* levels (median > 154 months) vs those with unaltered tumoral *ABAT* (median = 53 months; Figure 2a). Additionally, patients with upregulated *ABAT* had a nearly ten-fold increased progression-free survival estimate (median > 154 months) relative to those with unaltered *ABAT* levels (median = 16 months; Figure 2b). Unlike the Kaplan-Meier estimate, which compares samples with upregulated vs unaltered *ABAT* transcripts, Table 2 shows the relationship between *ABAT* transcript expression and additional clinical attributes; higher *ABAT* expression was significantly associated with disease-free status, living survival status, and no new neoplasms following the start of therapy (unpaired *t*-tests). Additionally, *ABAT* transcript levels were also significantly higher in the primary tumors of patients without metastases (*n* = 51; avg = 11.3 ± 0.3 log_2_ RSEM), relative to those of patients with metastases (*n* = 27; avg ± SEM = 9.0 ± 0.4 log_2_ RSEM) (unpaired *t*-test, *p* = 3.4 × 10^−5^). These findings are similar to the supplemental microarray data of a European ACC patient cohort, in which the expression of *ABAT* transcripts was higher in malignant tumors with favorable outcome, relative to those in patients with poor prognosis [18]. Similar to *ABAT*, *ALDH5A1* transcript expression was correlated with overall patient survival time (*n* = 78; Spearman *r* = 0.34; *p* = 2.4 × 10^−3^). Furthermore, patients whose tumors have upregulated *ALDH5A1* transcripts had significantly higher overall and progression-free Kaplan-Meier survival estimates than patients with downregulated tumoral *ALDH5A1* transcript levels (respectively, upregulated vs unaltered *n* = 10, 13; survival > 100 months vs < 22 months for each; log-rank test; *p* < 0.05). *GAD1* transcript levels were not correlated with clinical attributes. 

Tumors from ACC patients displayed no mutations or deep deletions in the *ABAT* gene. Two patient tumors displayed *ABAT* gene amplification (indicating high-level focal amplification), three a shallow deletion (heterozygous deletion), 29 (38%) of the tumors were diploid (no change), and 42 (55%) had a gain (low-level broad amplification), as defined by the Genomic Identification of Significant Targets in Cancer (GISTIC) algorithm. As expected, these CNAs were positively correlated with *ABAT* transcript expression (*n* = 76; Spearman *r* = 0.65; *p* = 2.7 × 10^−10^). Furthermore, tumors with the lowest all-gene mutational burden displayed the highest *ABAT* expression (*n* = 78; Spearman *r* = −0.54; *p* = 3.5 × 10^−7^). *ABAT* transcript expression level was negatively correlated with *ABAT* gene body methylation (*n* = 79; Spearman *r* = −0.50; *p* = 2.3 × 10^−6^). Favorable clinical attributes such as a living survival status and no new neoplasms following the start of therapy were associated with a lower degree of *ABAT* methylation (Table 3; unpaired *t*-tests). In seeking to identify the DNA methyltransferase (DNMT) putatively responsible for methylation of *ABAT*, we found that tumors with upregulated *DNMT1*, but no other DNMTs, expression displayed higher levels of *ABAT* methylation relative to those with unaltered *DNMT1* expression (average β-values ± standard deviation = 0.83 ± 0.14 vs 0.66 ± 0.24, respectively; unpaired *t*-test, Benjamini-Hochberg correction; *q* = 0.013). Additionally, upregulation of *DNMT1* and *ABAT* were mutually exclusive (log_2_ odds ratio < −3; *p* = 0.008), and *DNMT1* upregulation was negatively prognostic for ACC patient survival (Appendix A). 

Gene enrichment analysis of tumors with upregulated *ABAT* transcripts showed that 2029 genes were “underexpressed” and 2324 genes were “overexpressed” (Figure 3); the latter category included *GAD1*, which had median expression levels of 9.2 and 7.8 log_2_ RSEM in tumors with upregulated and unaltered *ABAT* transcripts, respectively. The most significantly underexpressed gene in samples with upregulated *ABAT* was *NAGS* (unpaired *t*-test, Benjamini-Hochberg correction; *q* = 1.4 × 10^−10^). The average transcript expression levels of *NAGS* indicated that it was present in samples with unaltered *ABAT* transcripts, while the average *NAGS* expression in samples with upregulated *ABAT* did not meet our 3.0 log_2_ RSEM cutoff (Table 4). This is particularly intriguing as *NAGS* encodes an enzyme, *N*-acetylglutamate synthase, that utilizes glutamate as a substrate. As the enzymes encoded by *NAGS* and *GAD1* may compete for glutamate as a substrate, the negative correlation of *NAGS* and *GAD1* transcript expression (*n* = 78; Spearman *r* = −0.34, *p* = 2.6 × 10^−3^) indicates a potential shift in glutamate utilization towards the GABA shunt in tumors with *ABAT* upregulation. In addition to *GAD1* and *NAGS*, other transcripts encoding GABA shunt-proximal enzymes of note included *AKR7A2*, *SDHA*, and *GLS* (encoding succinic semialdehyde reductase (SSR), succinate dehydrogenase complex flavoprotein subunit A (SDHA), and glutaminase (GLS), respectively) (Appendix A)—each of which displayed differential expression between tumors with upregulated vs unaltered *ABAT* (Figure 3; Table 4). To further explore the relationships between *ABAT* and the above-mentioned genes that were revealed by gene enrichment analysis, for each of the selected transcripts we examined both the 1) correlation of co-expression and 2) co-occurrence of upregulation with that of *ABAT*. Expression of each of the selected transcripts encoding GABA shunt-proximal genes was correlated with the expression of *ABAT* transcripts: for *NAGS*, *AKR7A2*, *SDHA*, and *GLS*, Spearman *r* = −0.75, −0.38, 0.45, and 0.37, respectively; *q* = 9.3 × 10^−12^, 3.8 × 10^−3^, 5.5 × 10^−4^, and 6.1 × 10^−3^, respectively (Appendix A). Upregulation of *SDHA* co-occurred with that of *ABAT*, while upregulation of *AKR7A2* was mutually exclusive with upregulation of *ABAT* (log_2_ odds ratio = 2.0, < −3 respectively; *q* = 0.035 for each). Additionally, because the GABA shunt bypasses two steps of the TCA cycle, we examined the expression of transcripts encoding the active subunits of the TCA cycle enzyme complexes that are bypassed by the GABA shunt. Unexpectedly, *ABAT* transcript expression was positively correlated with the expression of both *OGDH* (oxoglutarate dehydrogenase, OGDH) (Spearman *r* = 0.34; *q* = 0.013) and *SUCLG2* (succinate-CoA ligase [GDP-forming] subunit beta, mitochondrial, SUCLG2) (Spearman *r* = 0.38; *q* = 4.9 × 10^−3^). 

Through gene ontology analysis of 455 overexpressed genes and 264 underexpressed genes, we identified two biological pathways enriched in tumors with upregulated *ABAT*: “mesenchymal to epithelial transition” (MET) and “interferon gamma signaling” (Table 5)—both of which are related to decreased cancer progression [22,23]. Kaplan-Meier analysis on each of these 27 overexpressed genes revealed that patients whose tumors display upregulated *GPRC5A* and/or *PPL* (bolded, Table 5) transcripts had higher survival estimates than those with unaltered transcript levels (each with median survival of >154 months vs ~70 months; log-rank test; *p* < 5 × 10^−4^, Appendix A). Remarkably, no patients whose tumors expressed upregulated *PPL* (16 out of 78) were recorded as having died during the 154-month observation period. 

### 2.2. ABAT Transcripts and Corticosteroid Phenotype

Normal cortisol levels in ACC patients are positively prognostic, while patients with hypercortisolism fare less well [24]. Excess serum cortisol is present in 30–40% of ACC patients [6], which is consistent with the 45% (33 out of 74) in the TCGA dataset. Tumors of patients with normal serum cortisol levels displayed higher *ABAT* transcript expression than those with excess cortisol (11.7 ± 0.3 vs 8.9 ± 0.4 log_2_ RSEM respectively; unpaired *t*-test, *p* = 1.2 × 10^−6^) (Figure 4a), and are also over five times more likely to display upregulated levels of *ABAT* transcripts (57%, 27 out of 47) than tumors from patients with excess serum cortisol (11%, 4 out of 38). Among those with normal serum cortisol levels, Kaplan-Meier survival estimates showed that *ABAT* upregulation was favorable for survival (*z*-score ≥ 0.5; median survival for patients with upregulated *ABAT* transcripts > 154 months, unaltered *ABAT* = 45 months; log-rank test; *p* = 3.0 × 10^−3^), and approached significance even in patients with excess cortisol despite lower statistical power (data not shown). In this latter cohort, those with upregulated *ABAT* transcripts (*n* = 5) had a lower likelihood of developing new neoplasms following initial therapy, relative to unaltered (*n* = 28) (chi-squared test; *p* = 2.6 × 10^−3^). 

Enrichment analysis of ACC tumors with upregulated vs unaltered *ABAT* transcripts showed that *HSD3B2*, the gene encoding the steroidogenesis enzyme 3β-HSD (3-beta-hydroxysteroid dehydrogenase), was underexpressed in tumors with upregulated *ABAT* transcripts (Figure 4b, left panel), consistent with these tumors largely having originated from patients without excess cortisol. The overexpression of *NR3C1*, the gene encoding the glucocorticoid receptor (Figure 4b, right panel), was further identified by enrichment analysis. 

### 2.3. GABA_A_ Receptors

Transcripts encoding GABA_A_ receptor subunits displayed a wide range of expression among ACC tumors—with many tumors expressing transcript levels ≥ 3 log_2_ RSEM (Figure 5). This suggests that a variety of GABA_A_ receptors subtypes may be present in ACC tumors. Receptor subunits comprising heteropentameric GABA_A_ receptors are encoded by six *GABRA*_ genes (α1-6 subunit isoforms), three *GABRB*_ genes (β1-3 subunit isoforms), three *GABRG*_ genes (γ1-3 subunit isoforms), as well as *GABRD* (δ subunit), *GABRE* (ε subunit), GABRP (π subunit) and *GABRQ* (θ subunit) genes. Homopentameric GABA_A_ receptors are encoded by *GABRR*_ genes (ρ1-3 subunit isoforms) (formerly GABA_C_ receptors) and were not further considered here due to their low expression levels.

Of particular interest for their prognostic value and frequency of upregulation were two GABA_A_ receptor subunit-encoding genes: *GABRB2* and *GABRD*. *GABRB2* transcripts were the most highly expressed β subunit-encoding transcript and were positively correlated with *ABAT* transcript expression (Spearman *r* = 0.33; *q* = 0.02). Upregulation of *GABRB2* transcripts occurred in 36% of patient tumors (28 out of 78), and Kaplan-Meier survival analysis showed that *GABRB2* was positively correlated with a significantly higher estimated rate of overall survival (median > 154 months for patients with upregulated tumoral *GABRB2* vs 69 months for those with unaltered *GABRB2*) (Figure 6a) and a trend towards an increased estimate for progression-free survival (medians = 78 months and 25 months, respectively) (Figure 6b). Increased expression of *GABRB2* was correlated with several positive indicators of patient prognosis, including a lack of new neoplasm events following initial therapy (no new neoplasm: *n* = 33, avg ± SEM = 4.7 ± 0.42 log_2_ RSEM, new neoplasm: *n* = 35, avg ± SEM = 3.3 ± 0.4 log_2_ RSEM; unpaired *t*-test, *p* = 0.02). On the other hand, *GABRD* transcripts were present in nearly all ACC tumors, and were negatively correlated with *ABAT* transcript expression (Spearman *r* = −0.39; *q* = 3.6 × 10^−3^). *GABRD* transcripts were upregulated in 15% (12 out of 78) of ACC tumors, and this upregulation trends towards mutual exclusivity with *ABAT* upregulation (log_2_ odds ratio < −3; *p* = 0.064). Kaplan-Meier estimates showed that upregulation of tumoral *GABRD* transcripts was negatively prognostic for overall (Figure 6c) and progression-free survival (Figure 6d) in ACC patients relative to those patients with unaltered *GABRD*. Median survival of patients with upregulated vs unaltered tumoral *GABRD* transcripts was 18 vs 79 months for overall survival; 8 vs 78 months for progression-free survival. Increased expression of *GABRD* was correlated with negative indicators of patient prognosis, including a decrease in months of progression-free survival (*n* = 78; Spearman *r* = −0.36; *p* = 2.8 × 10^−3^) and an increase in the occurrence of new neoplasm events following initial therapy (no new neoplasm: *n* = 34, avg ± SEM = 5.9 ± 0.3 log_2_ RSEM; new neoplasm: *n* = 35, avg ± SEM = 7.2 ± 0.2 log_2_ RSEM; unpaired *t*-test, *p* = 8.6 × 10^−4^). Enrichment analysis of tumors with upregulated *GABRB2* or *GABRD* showed no biological pathway associations.

Because of *GABRB2*’s upregulation in over a third of ACC tumors and its positive association with patient clinical outcome, as well as the expression of other GABA_A_ receptor subunit-encoding genes, we examined whether the transcript complement of each tumor supports the expression of heteropentameric GABA_A_ receptors. The general stoichiometry of GABA_A_ receptors is two α subunits, two β subunits, and a “fifth-position” subunit—a composition that permits the formation of many distinct GABA_A_ receptor subtypes [25]. Over 75% of ACC tumors expressed either *GABRB2* or *GABRB3* transcripts, with approximately a third expressing both genes. Since the presence of a β subunit is necessary, although not sufficient, for the formation of GABA_A_ receptors, we next analyzed whether the 60 tumors expressing *GABRB2* and/or *GABRB3* genes also expressed transcripts encoding other subunits that are requisite for pentamer formation, i.e., at least one α subunit isoform and a fifth-position subunit (e.g., γ, δ, or ε). Twenty-three of these 60 tumors expressed transcripts encoding at least one of the α subunits, most commonly *GABRA3* or *GABRA5.* Of tumors expressing *GABRA3* or *GABRA5* transcripts, all expressed both *GABRD* and *GABRE*, while over half also expressed *GABRG1*. These findings suggested that some ACC tumors contain multiple subtypes of GABA_A_ receptors. Figure 7a visualizes the number of patient tumors expressing combinations of these various transcripts. Based on transcript analysis, the most abundant putative heteropentameric receptor configurations were permutations of α_3/5_β_2/3_γ_1_/δ/ε (Figure 7b). *GABRA5*, the most abundantly expressed α subunit-encoding gene, was upregulated in 10% of ACC tumors. *GABRA5* upregulation co-occurred with that of *GABRB3* (log_2_ odds ratio > 3; *q* = 0.003), the gene adjacent to *GABRA5* on chromosome 15 [26]. 

### 2.4. GABA_B_ Receptors

The GABA_B_ receptor is a heterodimeric G protein-coupled receptor composed of two subunits, B1 and B2 (encoded by *GABBR1* and *GABBR2*, respectively). As shown in Figure 8, 100% and 69% of ACC tumor samples expressed transcripts encoding *GABBR1* and *GABBR2*, with upregulation in 8% and 15% of patients, respectively. Although neither *GABBR1* nor *GABBR2* upregulation correlated with patient clinical outcome, their high expression levels indicated that GABA_B_ receptors were likely to be present in the majority of patient tumors. 

### 2.5. GABA Transporters

Plasma membrane GABA transporters (GAT-1, GAT-2, GAT-3, and BGT-1, encoded by *SLC6A1*, *SLC6A13*, *SLC6A11*, and *SLC6A12*, respectively) shuttle GABA between the cytoplasm and the extracellular environment. The vesicular GABA transporter (VGAT, encoded by *SLC32A1*) packages GABA into vesicles for coordinated release. Of these genes, *SLC6A1* transcripts were the most highly expressed in ACC tumors (avg = 5.5 ± 0.3 log_2_ RSEM; Figure 9), with 4% of tumors displaying upregulation. *SLC6A1* transcript expression was correlated with that of *ABAT* (Spearman *r* = 0.28; *q* = 0.049). Additionally, approximately half of tumors contained *SLC6A12* and/or *SLC6A13* transcripts, each averaging 3.2 ± 0.2 log_2_ RSEM, with 32% and 38% of tumors displaying upregulation, respectively. Upregulation of *SLC6A12* and *SLC6A13* transcripts significantly co-occurred in ACC tumors (log_2_ odds ratio >3; *q* < 0.001). Furthermore, enrichment analysis revealed 194 overexpressed and 16 underexpressed genes in tumors with upregulated *SLC6A13*, with overexpressed *SLC6A12* as the second-most significantly altered gene (Figure 10a). Average *SLC6A12* transcript expression levels ± standard deviation were 4.4 ± 1.3 and 2.9 ± 1.1 log_2_ RSEM in samples with upregulated and unaltered *SLC6A13*, respectively (Figure 10b). 

Although upregulation of GABA transporter-encoding genes was not correlated with patient survival, and no biological pathways were enriched in tumors with upregulated levels of these transcripts, expression levels indicate that GABA transporters are likely present in the majority of ACC tumors.

### 2.6. Differential Expression between ACC Tumors and Normal Adrenal Tissue

Due to the lack of normal adrenal samples in the TCGA dataset, we utilized two alternative methods of identifying genes differentially expressed between ACC tumors and normal adrenal tissue. The Gene Expression Profiling Interactive Analysis (GEPIA) user interface, which compares transcript expression level of TCGA data to that of the non-diseased adrenal gland (not specified as adrenal cortex) of deceased individuals in the Genotype-Tissue Expression (GTEx) database, revealed differential expression in 3 of the 30 GABA system genes. However, analysis of the National Center for Biotechnology Information (NCBI) gene set GSE10927, via the Oncomine™ user interface, identified 16 of 30 genes as differentially expressed. Common to both analyzed cohorts was the underexpression of *GABBR1*, and the overexpression of *GABRD*, in tumor vs normal tissues. *GABRD* is of particular interest, as it has been identified as a pan-cancer marker [27]. Due to the disparate outcomes between these sources, and as differential gene expression between tumor vs normal was not a focal point of our work, further analysis was not conducted. 

### 2.7. ACC Cell Line

NCI-H295R is a human ACC cell line with a steroidogenic phenotype, often used as a model for ACC [28]. A search of the NCBI’s Gene Expression Omnibus (GEO) identified two microarray datasets in which transcript data for NCI-H295R are available: GDS3537 and GDS3556. Of the 30 GABA-system transcripts queried, 10 had a rank expression level > 50% (as averaged across all probes, relative to expression level of all genes) in one or both studies: *GAD1*, *ABAT*, *ALDH5A1*, *GABRA3*, *GABRB3*, *GABRG1*, *GABRG2*, *GABBR1*, *GABBR2*, and *SLC6A13.* Our subsequent evaluation of GABA system transcript expression using RT-PCR revealed that NCI-H295R cells express transcripts encoding GABA shunt enzymes (*GAD1*, *ABAT* and *ALDH5A1*), GABA_A_ receptor subunits (*GABRA2*, *GABRA3*, *GABRA5*, *GABRB1*, *GABRB2*, *GABRB3*, *GABRG1*, *GABRG2*, *GABRG3*, *GABRD*, and *GABRE)*, both subunits of the GABA_B_ receptor (*GABBR1* and *GABBR2*), and all GABA transporters (*SLC6A1*, *SLC6A11*, *SLC6A12*, *SLC6A13*, and *SLC32A1*). A comparison between the GEO microarray/RT-PCR data vs TCGA tumor data found the common expression of many GABA system genes (Table 6). Of particular interest, *GABRD* (bold, Table 6) was highly present in ACC tumors, but not strongly expressed in the NCI-H295R cell line.

## 3. Discussion

This comprehensive bioinformatics assessment of the GABA system transcriptome in TCGA ACC patient tumors reveals that these tumors express transcripts encoding GABA shunt enzymes, multiple GABA receptor subunits, and GABA transporters. Of particular note, transcripts encoding GABA’s metabolizing enzyme, ABAT, are upregulated in more than 40% of ACC tumors, and its high expression is associated with favorable clinical attributes, including a three-fold increase in patient survival time. Tumors with upregulated *ABAT* transcript have enriched expression of genes in the gene ontology pathways of “mesenchymal-to-epithelial transition” and “interferon gamma signaling,” two pathways associated with limiting the spread of cancer. Our assessment of all GABA shunt transcripts, and those encoding other enzymes proximal to the shunt (such as those in the TCA cycle), implicates a GABA shunt metabolic phenotype that favors mitochondrial metabolism in patients with positive prognoses. Beyond these metabolic ramifications, ACC tumors express transcripts encoding multiple GABA transporters, suggesting that intracellular and intercellular tumoral GABA levels are highly regulated. Additionally, tumors express transcripts for both GABA_A_ and GABA_B_ receptor subunits. Regarding the former, upregulation of two transcripts encoding GABA_A_ receptor subunits, *GABRB2* and *GABRD*, are prognostic. This extensive GABA system transcriptome in ACC tumors is largely mirrored by the ACC cell line NCI-H295R, suggesting that this cell line is an appropriate in vitro model for studying the role of the GABA system in growth phenotypes for this cancer.

Approximately one-third of ACC patients have elevated serum cortisol levels [6], with symptomatic hypercortisolism negatively associated with ACC patient survival even following tumor resection and/or mitotane treatment [29]. For these patients, those few with upregulated tumoral *ABAT* transcripts trend toward positive clinical outcomes vs those with unaltered *ABAT* expression, while in patients with normal cortisol levels, upregulation of tumoral *ABAT* levels is strongly correlated with positive clinical outcome. Enrichment analysis of ACC tumors with upregulated *ABAT* identified two altered cortisol-related genes. Underexpression of *HSD3B2* (encoding 3β-hydroxysteroid dehydrogenase, 3β-HSD) was observed in tumors with upregulated *ABAT* transcript expression, i.e., those with favorable patient prognoses. Consistent with this, high prostatic 3β-HSD protein levels are correlated with several unfavorable clinical attributes in prostate cancer patients [30], while *HSD3B2* transcript expression has been proposed as a potential prognostic biomarker in lung squamous cell carcinoma [31]. The involvement of steroidogenesis pathways, especially those leading to cortisol production and glucocorticoid receptor activation, may be relevant to cancer as the glucocorticoid receptor can regulate the expression of hundreds of genes, including those affecting cell proliferation, apoptosis, and gene transcription [32,33]. In this regard, overexpression of *NR3C1* transcripts (encoding the glucocorticoid receptor) was noted in tumors with upregulated *ABAT* transcripts. Intriguingly, several glucocorticoid receptor binding sites are present within the *ABAT* gene (our mining of the Gene Transcription Regulation Database [34]), implying that the glucocorticoid receptor may regulate *ABAT* transcription. *ABAT*’s correlation with ACC patient hormone status and the expression of cortisol-related transcripts may represent an intersection of the GABA and steroid systems that could bear on ACC progression. 

In primary tumors of other cancers, high *ABAT* transcript expression is associated with favorable patient outcomes. Estrogen-receptor positive (ER^+^) breast cancers express highly upregulated levels of *ABAT* transcripts and its encoded protein, and have a more favorable patient prognosis, than low-*ABAT*-expressing hormone-insensitive breast cancers [35]. Consistent with these findings, low *ABAT* transcript levels are associated with resistance to endocrine therapy in ER^+^ breast cancer [36], as well as a decrease in relapse-free survival [35,37,38] and resistance to chemotherapy [39] in both ER^+^ and estrogen receptor negative (ER^−^) tumors. Notably, triple-negative breast cancer has relatively low levels of *ABAT* transcripts and protein, and a very poor patient prognosis [35], with overexpression of *ABAT* reducing migration and invasiveness in triple-negative cell lines [39]. Additionally, low levels of *ABAT* transcripts are associated with poor patient survival in mesenchymal glioblastoma [40] as well as progression, recurrence, and reduced patient survival time in hepatocellular carcinoma (HCC) [41]. In fact, ABAT protein is a highly sensitive biomarker for the identification of HCC and hepatoid adenocarcinomas [42]. Recently, high *ABAT* transcript levels were found to correlate with positive patient outcome in clear cell renal carcinoma, and to decrease cell proliferation and migration while increasing cell death in a kidney cancer cell line model [43]. High *ABAT* transcript expression is also favorably prognostic in renal (papillary), liver, and lung cancers (our mining of the Human Protein Atlas, v19.proteinatlas.org [44]), and prevalently expressed in other cancers, such as thyroid carcinoma [42], but is negatively prognostic for patients with esophageal squamous cell carcinoma [45]. 

From an epigenetic standpoint, *ABAT* gene-body methylation is negatively correlated with *ABAT* transcript expression in ACC, implying methylation-driven transcriptional silencing of this gene. Accordingly, as increased *ABAT* transcript expression is correlated with favorable ACC clinical outcome, increased levels of *ABAT* gene body methylation are associated with poor patient prognosis. It is possible that *ABAT* may be methylated by *DNMT1*, as *DNMT1* upregulation is associated with increased levels of *ABAT* methylation and is mutually exclusive with *ABAT* transcript upregulation. Although *DNMT3B* regulates *ABAT* methylation in noncancerous cells [46,47], we found no correlation between *DNMT3B* upregulation and *ABAT* methylation level in ACC tumors. In other cancers, reversal of transcriptional silencing by demethylating agents in a glioblastoma cell line increases *ABAT* transcript expression [48]. High levels of *ABAT* methylation are correlated with poor patient survival in myelodysplastic syndrome [49]. While the complexities of *ABAT* methylation remain to be understood, such regulation may impact cancer progression.

In general, the upregulation of *ABAT* transcripts suggests an increase in the expression of its encoded protein, the aminotransferase ABAT. If so, tumors of ACC patients with the most favorable prognoses would likely display an increase in GABA catabolism, the consequence of which affects two distinct processes, as depicted in Figure 11. Firstly, GABA activates GABA_A_ and GABA_B_ receptors. Therefore, GABA catabolism by ABAT lowers the amount of GABA available to activate GABA receptors. Secondly, the GABA shunt is a TCA cycle bypass mechanism beginning with the metabolism of glutamate to GABA by the activity of glutamic acid decarboxylase (GAD). Subsequently, the catabolic portion of the GABA shunt metabolizes GABA via ABAT activity into succinic semialdehyde (SSA), which is then metabolized by succinic semialdehyde dehydrogenase (SSADH) into succinate, a TCA cycle intermediate and substrate of the electron transport chain. The use of the GABA shunt to generate succinate has both anaplerotic and bioenergetic consequences. 

The upregulation of tumoral *ABAT* levels in a subset of primary ACC tumors reveals a prognostically favorable inter-patient metabolic heterogeneity among the ACC cohort that also includes the upregulation of *GAD1* and high expression of *ALDH5A1*. This “GABA shunt metabolic phenotype” is accompanied by transcript variations among genes encoding enzymes proximal to the GABA shunt that can affect the availability of glutamate for GABA synthesis. Focusing on this synthesis, co-expression of *ABAT* with *GLS*, which encodes the glutamine-to-glutamate aminohydrolase glutaminase, suggests that increased glutamate may be available for GABA synthesis in tumors with high levels of *ABAT* transcripts. In this context, *NAGS*, encoding *N*-acetylglutamate synthase (NAGS), is underexpressed in tumors with upregulated *ABAT*. As NAGS catalyzes the generation of *N*-acetylglutamate (NAG) from glutamate and acetyl-coA, decreased expression of *NAGS* transcripts may result in more glutamate available as a substrate for GAD67 (encoded by *GAD1*)—thus increasing GABA synthesis (Figure 11). These data support the existence of a GABA shunt-centric inter-tumoral metabolic heterogeneity among the ACC cohort. Interestingly, an intra-tumoral heterogeneity of GABA levels has recently been reported in clonal cell lines of distinct metabolic phenotypes (e.g., glycolytic vs mitochondrial) derived from single tumor ex vivo mouse pancreatic cancer cells [50]. Evaluation of both intra- and inter-tumoral heterogeneity of the GABA shunt may be important for understanding its role in mitochondrial metabolic heterogeneity, a feature of tumors that is increasingly recognized to influence cancer growth, progression, and drug treatment sensitivity [51].

Focusing on the catabolic arm of the GABA shunt, high *ALDH5A1* transcript expression is present in nearly all primary ACC tumors. This gene encodes SSADH, which converts SSA to succinate, ensuring that the GABA shunt is driven to completion. Succinate is a substrate of succinate dehydrogenase (SDH), a TCA cycle enzyme and the electron transport chain’s complex II. The catalytically active subunit of SDH (SDHA) is encoded by the tumor-suppressor gene *SDHA* [52], and converts succinate into fumarate while generating a high-energy reducing equivalent. In ACC tumors, the co-occurrence of *SDHA* and *ABAT* upregulation supports the notion that GABA shunt activation contributes to succinate production. Additionally, the mutual exclusivity of *ABAT* upregulation with that of *AKR7A2* (which encodes succinic semialdehyde reductase (SSR) and may lead to the diversion of GABA carbons out of the GABA shunt), and the under-expression of *AKR7A2* in ACC tumors with upregulated *ABAT* transcripts may further ensure that the GABA shunt in these tumors culminates with the production of succinate. Unsurprisingly, *AKR7A2* upregulation is negatively prognostic for disease-free and progression-free survival, compared to patients with downregulated *AKR7A2* (unpublished). As the GABA shunt is commonly thought to act as a diversionary bypass of two steps of the TCA cycle, we were surprised to discover that expression of both *OGDH* and *SUCLG2* transcripts, which encode the active subunits of the enzymes in these bypassed steps, were positively correlated with that of *ABAT*, potentially indicating an increase in succinate production by both pathways. As both the TCA cycle and the GABA shunt may serve anaplerotic and bioenergetic purposes, these data imply an enhanced mitochondrial metabolism in a subset of primary ACC tumors in patients with favorable prognoses. 

Metastases contribute to most cancer-related deaths [53]. Our analysis of ACC patient clinical attributes revealed that higher levels of *ABAT* transcript expression are correlated with lower-stage, organ-confined cases of ACC. In keeping with this finding, upregulated *ABAT* expression is correlated with enriched expression of genes in the “mesenchymal to epithelial transition” (MET) pathway, which opposes the epithelial to mesenchymal transition (EMT) required for metastatic spread [23]. Indeed, induction of the MET pathway in cancer cell lines reduces proliferation, migration, and/or invasion in vitro, as well as tumor growth and/or metastatic capability in mouse models [54,55,56]. Of the MET pathway genes identified here, the majority are experimentally associated with maintaining an epithelial phenotype in cancer cells [57,58,59,60] and/or cell adhesion complexes that prevent migration and invasion [61,62,63], or are known to be downregulated during EMT [61,64,65]. Strikingly, when we evaluated the twenty enriched MET genes for their prognostic value in ACC patient outcome, expression of two genes—*PPL* (a structural component of desmosomes) and *GPRC5A* (a G protein-coupled receptor with an unidentified endogenous ligand)—were found to be associated with increased patient survival. In fact, no patients whose tumors demonstrated upregulated *PPL* died during the 154-month observation period, whereas those with lower *PPL* expression had a median survival time of 69 months. 

The enrichment of MET pathway genes in tumors with upregulated *ABAT* transcripts indicates a decreased likelihood of invasive cells exiting the primary tumor, suggesting that low *ABAT* levels are associated with a predisposition for metastasis. Consistent with this notion, as shown here, primary ACC tumors in patients with metastases have significantly lower *ABAT* transcript levels than primary tumors without metastases. This is in keeping with the correlation of low *ABAT* transcript expression with both an increased risk of metastasis in triple-negative breast cancer tumors [39] and an increased expression of a “metastasis gene signature” in hepatocellular carcinoma [41]. While low *ABAT* transcript levels may be important for EMT and the initiation of metastasis, the colonization of the metastatic niche requires activation of the MET pathway [23]. Therefore, we predict that, in ACC metastatic spread, secondary-site colonization is associated with an increase in *ABAT* transcript levels. Consistent with this prediction, both human breast-to-brain metastases [66], and brain metastases of various cancers in xenograft mouse models [67] show elevated *ABAT* transcript expression relative to primary tumors. In fact, the ABAT inhibitor vigabatrin suppressed the proliferation and the number of brain metastases, and reduced the size of existing brain metastases, in xenograft mouse models [67]. In humans, breast-to-brain metastases also express increased levels of *GAD1* transcripts [66], while brain metastasis of ccRCC contain high levels of GAD67 protein [68]. Mouse models of brain metastases similarly show elevated levels of *GAD1* transcripts [67].

*ABAT* encodes a dual-function mitochondrial enzyme responsible for both GABA catabolism and maintenance of mitochondrial DNA copy number [69]. Other genes encoding mitochondrial proteins that are prognostic for survival in ACC include *PINK1*, encoding a mitochondrial serine/threonine kinase [18,70], and *FATE1*, encoding a mitochondrial fission factor-family protein [71,72]. Although not tested in ACC clinical trials in the United States (ClinicalTrials.gov), drugs targeting TCA cycle-associated enzymes are now in clinical trials for various other cancers [73]. We determined that many transcripts encoding these enzymes are prognostic for ACC outcome (unpublished). In support of mitochondrial metabolism as an ACC therapeutic target, a drug blocking such metabolism in a xenograft mouse model of ACC inhibits the growth of tumors without overt toxicity [74]. 

Our enrichment analysis also reveals that ACC tumors with upregulated *ABAT* transcripts are enriched for transcripts of the “interferon gamma (IFN-γ) signaling” biological pathway [75]. These enriched IFN-γ pathway genes are associated with decreased tumor progression and improved patient prognosis [76,77,78,79,80,81]. Of particular note are transcripts encoding HLA complex proteins (Table 5), which are commonly associated with tumor regression [82], as the expression of HLA proteins on the surface of cancer cells allows T-cells to recognize and destroy cancer cells [83]. Interestingly, T-cells synthesize [84] and secrete [85] GABA, and express multiple components of the GABA system—including *GAD1* and *ABAT* transcripts, as well as functional GABA transporters, which may be upregulated upon T-cell activation [84]. Additionally, GABA_A_ receptor activation suppresses T-cell proliferation [84,86,87]. 

ACC tumors express transcripts encoding three distinct plasma membrane GABA transporters—*SLC6A1*, *SLC6A12*, and *SLC6A13*. Importantly, the expression of *SLC6A1* transcripts in most ACC tumors is both high and positively correlated with *ABAT* transcript expression. Plasma membrane transporters in tumors may permit the exchange of GABA among tumoral cells, controlling GABA levels through bidirectional GABA transport as has been noted in both neurons [88] and cultured T-cells [85,89]. In this context, extracellular GABA can be taken up either for use as a metabolic substrate and/or to terminate receptor activation, as in the central nervous system. Conversely, intracellular GABA may be released into the extracellular tumor microenvironment (TME) to activate GABA receptors, or for cooperative metabolism between cells of the tumor. Such exchange of metabolites between tumor cells has been shown for other metabolic substrates [90,91]. 

Despite GABA being primarily recognized as a neurotransmitter, its broader importance is evidenced by the use of the GABA shunt by bacteria, yeast, and plants—which utilize the GABA shunt to mitigate stressors including oxidative, pH, and nutrient deprivation stress [92,93,94,95,96,97,98,99,100,101,102]. In particular, flux of carbons through the GABA shunt may affect ROS production, as supported by the demonstration that altered *ABAT* expression and ABAT activity affects mitochondrial oxygen consumption in non-cancerous cells [47,103], and that the deletion of *ABAT* orthologs in fungi causes increased sensitivity to oxidative stress [96]. Modulation of ROS production by GABA shunt activity, to our knowledge, is unexplored with respect to cancer. 

The presence of both GABA_A_ and GABA_B_ receptor-encoding transcripts in ACC tumors suggests that both types of GABA receptors may be operable in ACC. With respect to the GABA_A_ receptor, *GABRB2* and *GABRD*, the genes encoding the receptor’s β2 and δ subunits, are both highly expressed in ACC tumors, and are positively and negatively prognostic for ACC patient survival, respectively. The most commonly contemporaneously-expressed transcripts within individual tumors are *GABRA3*, *GABRA5*, *GABRB2*, *GABRB3*, *GABRG1*, *GABRD*, and *GABRE*—encoding the α3, α5, β2, β3, γ1, δ, and ε subunits, respectively—with the vast majority of all ACC tumors expressing transcripts encoding both δ and ε subunits. Given that the prototypic GABA_A_ receptor composition requires two α subunits, two β subunits, and a fifth position subunit (γ, δ, ε, θ, π), we propose that the most common putative pentameric receptor schemas found in ACC tumors are permutations of α_3_α_5_β_2_β_3_γ_1_/δ/ε receptors (Figure 7). Receptors composed of these subunits have been identified as being tentatively extant in brain [104], while similar receptor subunit compositions have been proposed in lung [105]. 

Correlation of GABA_A_ receptor subunit gene expression with cancer growth phenotypes in cancer cell lines, animal models, and patients strongly implicates GABA_A_ receptors in cancer progression. Expression of *GABRG3*, *GABRQ*, *GABRE*, and *GABRR2* is correlated with inhibition of growth phenotypes in cell lines and with favorable patient outcomes [106,107,108] (Broad Institute of MIT & Harvard, firebrowse.org), while expression of other transcripts (*GABRB2*, *GABRP*) is associated with cancer progression [109,110,111]. Of particular relevance to our work, high expression of *GABRA3* transcripts, or its encoded α3 protein, is correlated with poor patient outcome in neuroblastoma [112] and lung adenocarcinoma [113,114,115]. In breast cancer cell culture and xenograft models, *GABRA3* transcripts undergo adenosine to inosine (A-to-I) RNA editing that decreases cell invasiveness vs unedited *GABRA3*, by inducing intracellular retention of α3 subunit-containing receptors [116]. Although *GABRA3* transcripts are not prognostic for ACC patients, their A-to-I editing status in ACC tumors has not been determined. 

Pharmacological experiments with GABA_A_ receptor ligands demonstrate that receptor activation induces cell death in ex vivo medulloblastoma cells [117] and a transgenic mouse model of neuroblastoma [112], and inhibits the invasiveness and tumor formation of colon carcinoma cells in vitro and in xenograft mice [118,119]; while blocking GABA_A_ receptors promotes tumor growth of glioblastoma [120]. Not all studies, however, have found GABA_A_ receptor activation to inhibit cancer cell growth, as receptor activation increases cell proliferation in prostate adenocarcinoma and oral squamous cell carcinoma cell lines [121,122,123]. Such discrepant findings between studies of different cancers may be due to the varying composition of heteropentameric GABA_A_ receptors, as the 19 distinct subunits impart different functional characteristics (reviewed in [104]). 

In addition to mediating effects via GABA_A_ receptors, GABA also activates the GABA_B_ receptor, a heterodimeric G protein-coupled receptor. While two-thirds of ACC patient tumors express transcripts encoding both GABA_B_ receptor subunits, neither are correlated with patient prognosis. However, it is likely that GABA_B_ receptors are functional in ACC tumors, as shown in the ACC NCI-H295R cell line—albeit with undetermined effects on cell growth phenotype [20]. In other cancers, activation of GABA_B_ receptors enhances the invasiveness of prostate adenocarcinoma cell lines [124,125,126], while inhibiting migration and proliferation in cell lines of hepatocellular carcinoma [127,128], pancreatic cancer [129], and lung adenocarcinoma [115,130], as well as decreasing tumor volume in hepatocellular carcinoma mouse xenografts [127]. Given the ability of GABA_B_ receptors to affect growth phenotypes in other cancers, the presence of their transcripts in ACC tumors, and their functionality in the NCI-H295R ACC cell line, the impact of GABA_B_ receptors on cancer growth phenotypes warrants investigation. 

The NCI-H295R cell line is a widely-used [28,131], and by far the most widely-cited (>800 citations in PubMed) ACC cell line model, that we show has a strikingly similar GABA system transcriptome to that of ACC tumors. Such similarity indicates that GABA system components likely reside, at a minimum, within the cancerous cells of the tumor. Like ACC tumors, the NCI-H295R cell line expresses transcripts encoding GABA shunt enzymes as well as a variety of GABA receptor subunits and multiple GABA transporters. In fact, these cells express *GAD1* transcripts that encode functional GAD67 protein, as determined by the production of ^14^C-CO_2_ from ^14^C-glutamate [20]. Of potential relevance to cancer, treatment of these cells with endocrine disrupting agents induces a four-fold upregulation of *GAD1* transcript levels [132] (see the supplementary microarray data in Ref. 132) and a significant decrease in *ABAT* transcript levels (unpublished, determined using Song, et al.’s methodology). Due to the similar GABA system transcriptomes between ACC tumors and the NCI-H295R cell line, and the ability of endocrine disruptors to affect this transcriptome, this cell line is a particularly appropriate in vitro ACC model for assessing the role of the GABA system in cancer cell growth phenotypes such as proliferation, migration, and invasion. 

One notable difference between ACC tumors and the NCI-H295R cell line is that ACC tumors display high levels of *GABRD* transcript expression, while we observed only relatively low levels expressed in the cell line. One explanation for this disparity is that *GABRD* transcripts may be predominantly expressed in ACC tumor host cells, such as T-cells. This notion is consistent with an unsupervised bioinformatics study of TCGA tumors (not including ACC) that identified *GABRD* expression as a pan-cancer marker that is overexpressed in tumor vs adjacent normal tissue **[27]**. Our data mining effort extends this Gross et al., finding to include ACC tumors. Additionally, our results indicate that within the ACC cohort, *GABRD* expression is upregulated in a subset of patient tumors with lower *ABAT* transcript levels, and that *GABRD* upregulation is not only unfavorably prognostic, but also may be associated with a decreased capacity to degrade GABA.

Our work identifies the expression levels and prognostic implications for transcripts encoding components of the GABA system—most notably, the GABA shunt and certain GABA_A_ receptor subunits, which largely appear together in the same subset of ACC tumors. Additionally, transcripts encoding GABA_B_ receptors and GABA transporters are ubiquitously present in ACC tumors. The use of human tumor data lends particular relevance to our findings. Beyond this strength, we present a full assessment of these genes as a system with respect to cancer, giving a context to the many unsupervised studies in which these genes have been individually identified. A limitation of this study is that bioinformatics data are correlational, and thus functional studies are required to determine whether any differences in transcript expression are biologically or phenotypically meaningful [133].

Future functional studies will need to be conducted in model systems to determine whether GABA shunt function and/or GABA receptor activation affects cell growth phenotypes or cell metabolism. To this end, we have determined that the NCI-H295R cell line GABA system transcriptome mirrors that of ACC tumors. Future studies of this cell line should confirm the presence of GABA system proteins and evaluate whether pharmacological and/or genetic manipulation of this system affects cell proliferation, migration, or invasion. Additionally, the metabolic phenotype associated with GABA’s role as a potential anaplerotic substrate should be investigated with respect to mitochondrial metabolism and bioenergetics. If functional studies in cell lines determine that manipulation of the GABA system impacts cell growth and metabolism, in vivo ACC models can be used to characterize whether the arsenal of GABA system therapeutics—currently used to treat neurological and psychiatric disorders [134,135]—affects tumor formation, growth and metastasis. Such therapeutics include those that inhibit ABAT (Sabril**^®^**), allosterically potentiate GABA_A_ receptors (Klonipin**^®^**, Solfoton**^®^**, and Brexanolone**^®^**, among many others), activate GABA_B_ receptors (Lioresal**^®^**), and inhibit GABA transporters (Gabitril**^®^**). If functional studies prove fruitful, the clinical availability of these pharmacological agents may offer an accelerated path towards drug discovery for ACC treatment. 

## 4. Materials and Methods 

### 4.1. TCGA ACC Data

We utilized v3.4.9 and prior versions (from January—September of 2020) of the cBioPortal user interface (cbioportal.org; [136,137] to access ACC data generated by the TCGA Pan-Cancer initiative [138,139]. Gene mutations, copy number alterations (CNA), mRNA transcript levels, and clinical attributes associated with the 30 GABA-system genes (Table 1) were evaluated within the TCGA Pan-Cancer dataset. All tumor samples analyzed originated from the adrenal gland (“tumor disease anatomical site”), i.e., primary tumor data. Each sample was from a distinct patient (demographic data in Appendix A). Mutation and CNA data were available for 89 patient tumor samples, while RNA-Seq V2 data were available for 78 samples. Since the TCGA Pan-Cancer dataset did not provide DNA methylation status, the Adrenocortical Carcinoma TCGA Firehose Legacy dataset (accessed via cBioPortal) was used to evaluate this parameter. Methylation data collected using Illumina Infinium Human Methylation 450K BeadChip arrays were available for 80 patients and were presented as beta-values (β), an intensity ratio representing methylated vs unmethylated alleles. In the event of the use of multiple probes per gene, data obtained by the probe with the strongest negative correlation between methylation and transcript expression was used. Additionally, the TCGA Firehose Legacy dataset was utilized to evaluate the clinical attribute “excess adrenal hormone history type” as well as correlations between levels of *ABAT* transcripts, serum cortisol, and transcripts encoding steroidogenic enzymes and the glucocorticoid receptor. It should be noted that patient sample data contained within the two ACC TCGA datasets used here largely overlap. 

Transcript data are presented as “RSEM,” indicating processing using the RNA-Seq by Estimation Maximization (RSEM) algorithm to estimate the maximum likelihood abundance for each gene [140]. A threshold log_2_ RSEM cutoff ≥ 3 was chosen to indicate presumed biologically relevant levels of GABA system transcript expression using GAD1 expression levels as a reference. GAD1 was chosen for three reasons: Firstly, *GAD1* transcript expression levels in the ACC TCGA Pan-Cancer dataset were similar to those of five genes identified as having “excellent diagnostic accuracy” for distinguishing malignant from benign ACC tumors [141]. The collective average expression level of these genes (*IL13RA2*, *HTR2B*, *CCNB2*, *RRARES2*, and *SLC16A9*) ranged from 5.5–11.8 log_2_ RSEM (average = 8.0 log_2_ RSEM), while the *GAD1* transcript expression average was 8.0 log_2_ RSEM. Secondly, *GAD1*, which encodes the GABA-synthetic enzyme glutamic acid decarboxylase 67 (GAD67), begins the reaction pathway in the production of GABA, and therefore its expression would be expected to affect GABA-dependent processes. Finally, GAD67 activity is present in the ACC cell line NCI-H295R [20]. Ultimately, our cutoff encompassed the *GAD1* expression level in 95% of ACC samples. In addition to evaluating average and median log_2_ RSEM expression levels for each GABA system gene, we examined among these genes their co-expression with each other as well as with additional glucocorticoid-related transcripts. Transcript “upregulation” of GABA system genes was assessed as an RSEM transcript value greater than one standard deviation (*z*-score cutoff) away from the average expression of the queried gene in all diploid samples, except where otherwise noted. Samples not expressing upregulated levels of a transcript are termed “unaltered.” Where necessary, cBioPortal Onco Query Language was used to specify subgroups of samples. For genes demonstrating upregulation in over a third of patient samples, gene enrichment analysis was performed and volcano plots are shown. Upregulated genes were further evaluated for their co-occurrence of upregulation within each tumor. 

Lastly, we examined the relationship of GABA system gene expression to clinical attributes. Kaplan-Meier estimates of overall and disease/progression-free survival were generated and analyzed by log-rank nonparametric test within the cBioPortal user interface. Patient survival data was available for 154 months following diagnosis. Clinical attribute data (defined by American Joint Committee on Cancer codes), including excess cortisol, were downloaded and analyzed for comparison against transcript level. Comparison of transcript levels in ACC primary tumors of TCGA patients with metastases vs those without was accessed via the Human Cancer Metastasis Database [142]

### 4.2. Gene Ontology

FunRich v3.1.3, a functional enrichment analysis tool (funrich.org; [143]), was used to determine the molecular functions and biological pathways associated with genes whose expression was enriched in samples with upregulated GABA system transcript levels. This program utilized gene ontology annotations compiled from various databases such as the Human Protein Reference Database, Entrez Gene, and UniProt. To identify altered gene categories, the average transcript value (log_2_ RSEM) for each identified “overexpressed” or “underexpressed” gene in the enrichment set with a log-fold cutoff ≥ |1.0| (455 and 264 genes, respectively) was compared against the program’s curated human database. Categories with Bonferroni-corrected *p*-values ≤ 0.05 were considered significantly altered. 

### 4.3. Transcript Expression in Tumor vs. Normal Tissues

As the TCGA tumor dataset did not contain adjacent paired normal controls for ACC patients (Broad Institute of MIT & Harvard, firebrowse.org), we compared TCGA ACC transcript expression data (*n* = 77) to data from the Genotype-Tissue Expression (GTEx) project, which contains “normal” tissue-specific gene expression data gathered from non-diseased adrenal gland (adrenal cortex not specified) of deceased individuals (*n* = 128), using the Gene Expression Profiling Interactive Analysis (GEPIA) user interface (gepia.cancer-pku.cn; [144]). TCGA and GTEx RNA-Seq raw data as provided by GEPIA were previously recomputed for standardization by the University of California Santa Cruz Xena project. Statistical analysis of differential gene expression between the transcript levels (log_2_ transformations of [transcripts per million + 1]) of TCGA vs GTEx samples was performed by the GEPIA portal, which defined thresholds as a log_2_ fold-change cutoff of median_tumor_—median_normal_ = 1.0 and a *p*-value cutoff of 0.01. Additionally, microarray data (Affymetrix Human Genome U133 Plus 2.0 Array) from the National Center for Biotechnology Information (NCBI)’s curated Gene Expression Omnibus (GEO) repository was accessed via Oncomine™ Research Edition (oncomine.org; ThermoFisher Scientific) to compare transcript expression data of 33 ACC tumors and 10 adrenal cortex surgical specimens from GSE10927 [145].

### 4.4. NCI-H295R Cell Line

The ACC cell line NCI-H295R [131] (CRL-2128, American Type Culture Collection, Manassas, VA, USA) was cultured in phenol red-free Dulbecco’s Modified Eagle Medium/Ham’s Nutrient Mixture F-12 supplemented with 2.5% Corning™ Nu-Serum I, 1% Corning™ ITS+ premix, and 1% Gibco™ penicillin/streptomycin (Thermo Fisher Scientific, Waltham, MA, USA) at 37^o^C and 5% CO_2_. Cells were harvested for RNA extraction using the E.Z.*N*.A.^®^ Total RNA Kit I (Omega Bio-Tek, Norcross, GA, USA), followed by cDNA synthesis using the iScript™ cDNA Synthesis Kit (Bio-Rad, Hercules, CA, USA). RT-PCR was performed with exon-spanning primers designed using Integrated DNA Technologies’ PrimerQuest^®^ tool (primer list in Appendix A). Quantitative real-time polymerase chain reaction (RT-PCR) was performed using iTaq Universal SYBR Green Supermix (Bio-Rad) on a Bio-Rad CFX96 Fast Real-Time PCR System and analyzed with CFX Manager software. Transcripts with threshold cycle (Ct) values lower than 35 cycles were categorized as “expressed” [146]. 

For additional evaluation of GABA system gene expression in the NCI-H295R cell line, we queried the NCBI’s GEO repository using the Boolean search term “H295R AND (GAD1 OR GAD2 OR ABAT OR ALDH5A1 OR GABRA1 OR GABRA2 OR GABRA3 OR GABRA4 OR GABRA5 OR GABRA6 OR GABRB1 OR GABRB2 OR GABRB3 OR GABRG1 OR GABRG2 OR GABRG3 OR GABRD OR GABRE OR GABRP OR GABRQ OR GABRR1 OR GABRR2 OR GABRR3 OR GABBR1 OR GABBR2 OR SLC6A1 OR SLC6A11 OR SLC6A12 OR SLC6A13 OR SLC32A1).” This search yielded results from two microarray (Affymetrix Human Genome U133 Plus 2.0 Array) studies: dataset GDS3537 (series GSE8588) and dataset GDS3556 (series GSE15918). Transcript data for other ACC cell lines was not available. 

### 4.5. Data Analysis 

In evaluating correlations between gene expression and patient clinical attributes, statistics generated by the cBioPortal user interface were utilized where available (*q*-values generated by Benjamini-Hochberg procedure to adjust for false discovery rate; corrected among >19,000 profiled genes). Where statistical analysis was not provided by the user interface, data were downloaded as .txt files, samples with RSEM = 0 were excluded, and unpaired *t*-tests were performed with GraphPad Prism version 8.4.1 for Windows (GraphPad Software, San Diego, CA, USA; www.graphpad.com). To adjust for the increased family-wise error inherent in performing multiple *t*-tests on the same samples, corrected *p*-values were generated by multiplying the calculated *p*-values by the number of *t*-tests performed on a given set of samples. Two-tailed *p*-values were used for all *t*-test calculations except those regarding methylation data, which utilized one-tailed *p*-values based on a unidirectional hypothesis. 

## Figures and Tables

**Figure 1 ijms-21-08485-f001:**
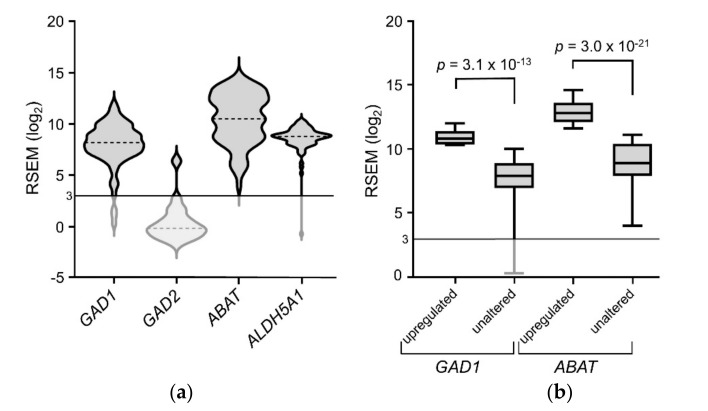
Expression level of transcripts encoding enzymes of the GABA shunt in ACC tumors. (**a**) Volcano plots depicting GABA shunt gene expression; dotted lines represent median expression levels. (**b**) Box-and-whisker plots representing expression of *GAD1* and *ABAT* transcript levels in tumor samples, upregulated vs unaltered, unpaired *t*-tests.

**Figure 2 ijms-21-08485-f002:**
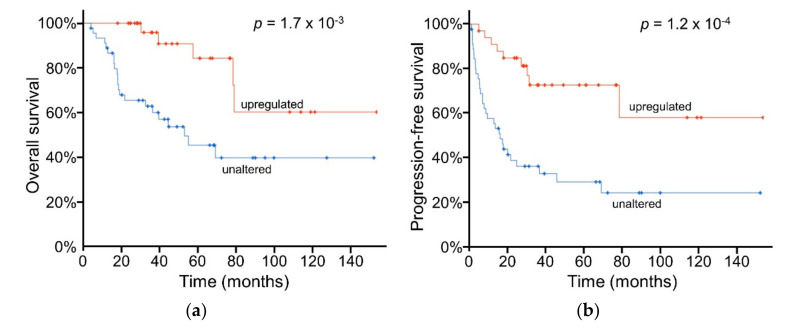
Kaplan-Meier survival estimates of patients with upregulated vs unaltered tumoral *ABAT* transcript expression. (**a**) overall survival and (**b**) progression-free survival in patients with tumors expressing upregulated *ABAT* transcripts (*n* = 33); relative to unaltered *ABAT* transcripts (*n* = 45), log-rank tests. Plots generated by cBioPortal, modified.

**Figure 3 ijms-21-08485-f003:**
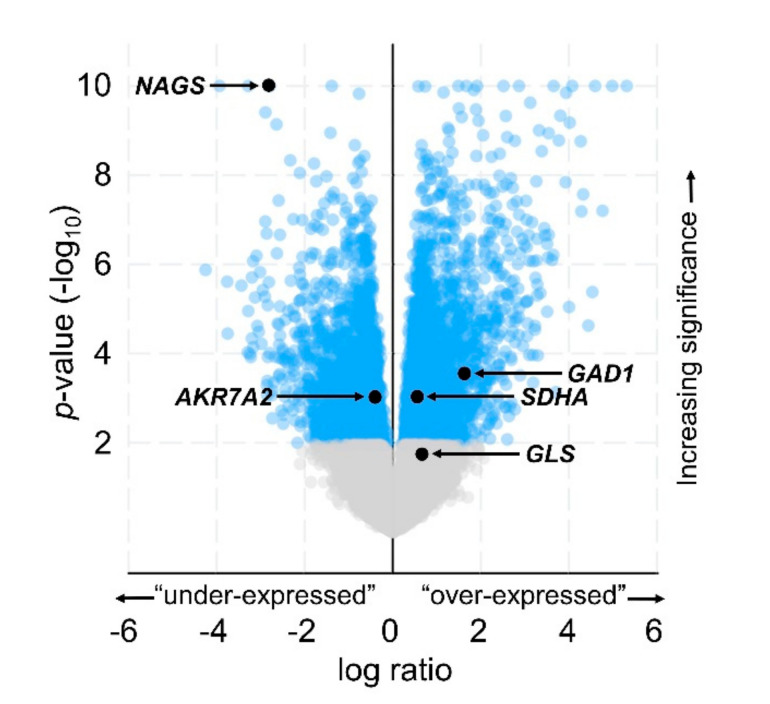
Transcripts altered in samples with upregulated *ABAT* transcripts. Volcano plot representing log_2_ transformation of the ratio of an enriched gene’s average transcript level in the *ABAT* “upregulated” samples relative to that gene’s average transcript level in the *ABAT* “unaltered” group of samples (*x*-axis = “log ratio”), as plotted against the *p*-value significance (Unpaired *t*-test) of each transcript’s enrichment in the “upregulated” samples (*y*-axis = log_10_
*p*-value). Gray = *q* > 0.05; blue = *q* ≤ 0.05 (Benjamini-Hochberg correction). Generated by cBioPortal, modified.

**Figure 4 ijms-21-08485-f004:**
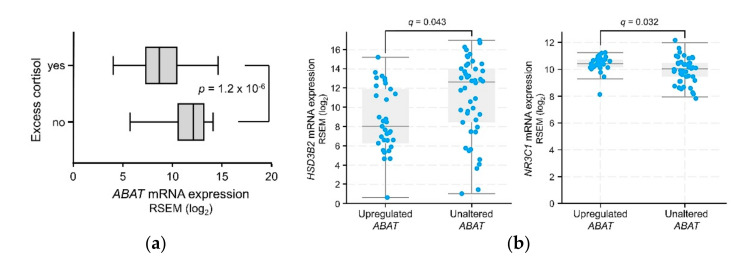
Relationships between *ABAT* transcript levels and steroid phenotype of ACC tumors. (**a**) Box-and-whisker plot of *ABAT* transcript expression levels in ACC patient tumors, separated by Excess Adrenal Hormone History Type (TCGA Firehose Legacy); dotted lines represent median expression levels. “Yes” *n* = 33; “No” *n* = 41. (**b**) Expression levels of *HSD3B2* and *NR3C1*, in tumors with upregulated vs unaltered *ABAT* transcript expression. (TCGA Firehose Legacy dataset). Unpaired *t*-tests, Benjamini-Hochberg correction; generated by cBioPortal, modified.

**Figure 5 ijms-21-08485-f005:**
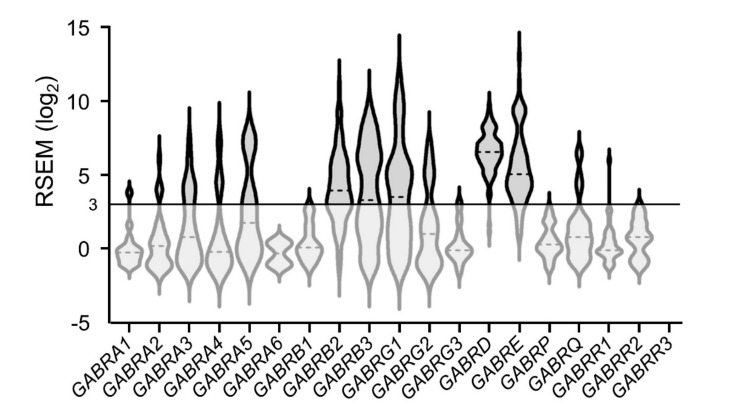
Expression of transcripts encoding GABA_A_ receptor subunits; dotted lines represent median expression levels. Gene names given on the *x*-axis.

**Figure 6 ijms-21-08485-f006:**
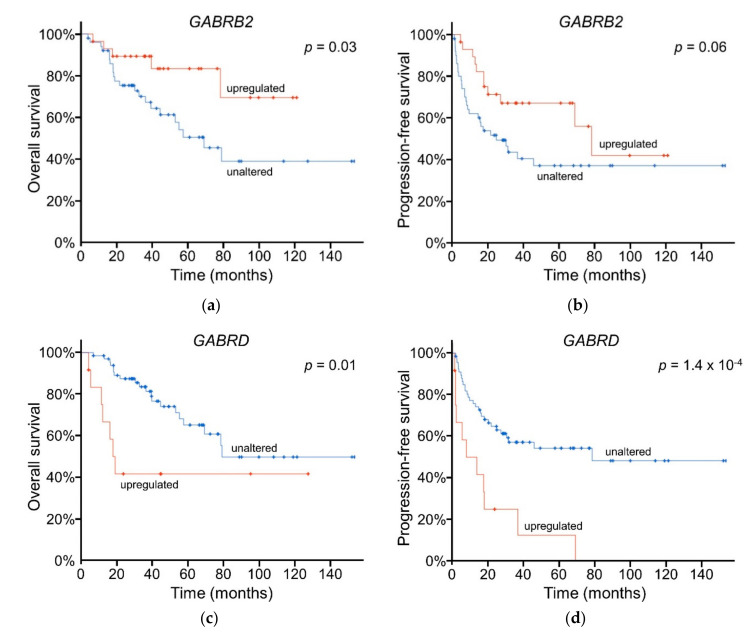
Kaplan-Meier survival estimates of patients with upregulated vs unaltered tumoral *GABRB2* or *GABRD* transcript expression. (**a**) overall survival and (**b**) progression-free survival in samples with upregulated *GABRB2* expression (*n* = 28) vs unaltered (*n* = 50). (**c**) overall survival and (**d**) progression-free survival in patients with tumors expressing upregulated *GABRD* transcripts (*n* = 12) vs unaltered (*n* = 66). Log-rank tests. Plots generated by cBioPortal, modified.

**Figure 7 ijms-21-08485-f007:**
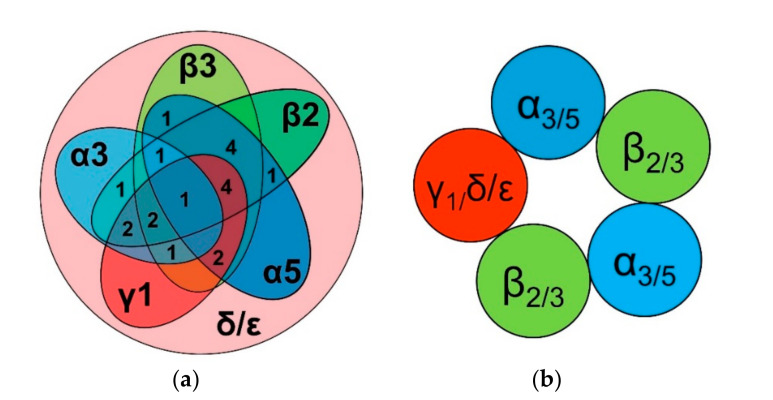
Overlap of expression of transcripts encoding GABA_A_ receptor subunits within individual ACC tumors. (**a**) Venn diagram representing the number of potential receptor-expressing tumors with >3 log_2_ RSEM transcripts encoding the indicated subunits. (**b**) Depiction of potential pentameric GABA_A_ receptor subunit configurations based on most prevalent subunit-encoding transcripts.

**Figure 8 ijms-21-08485-f008:**
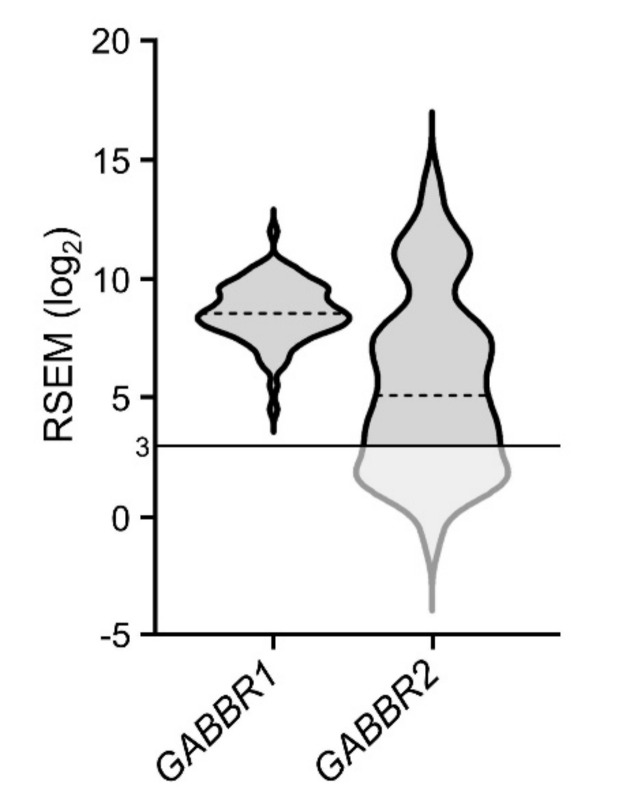
Expression of transcripts encoding GABA_B_ receptor subunits; dotted lines represent median expression levels.

**Figure 9 ijms-21-08485-f009:**
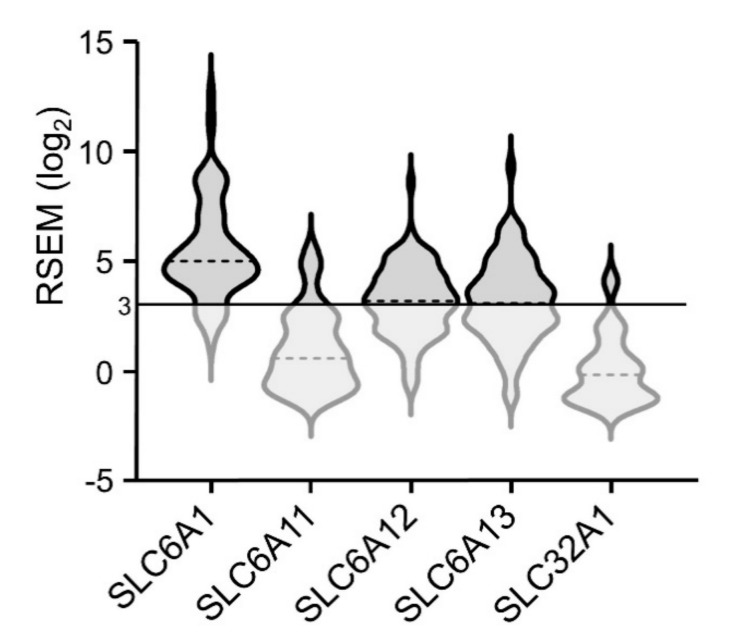
Expression of transcripts encoding GABA transporters; dotted lines represent median expression levels.

**Figure 10 ijms-21-08485-f010:**
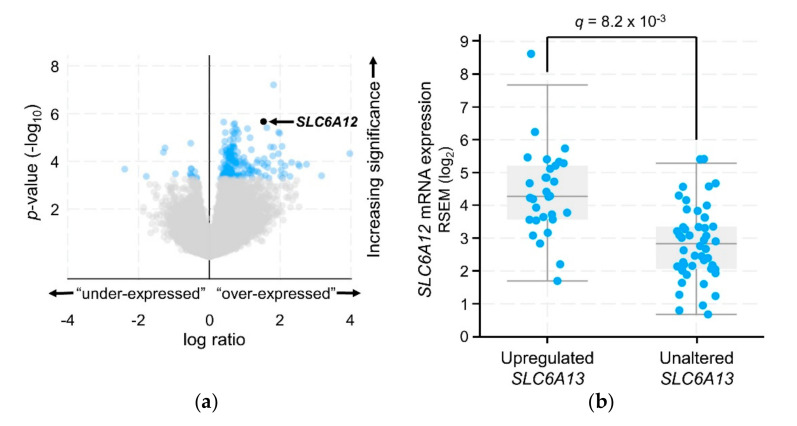
Genes altered in samples with upregulated *SLC6A13* transcripts. (**a**) Volcano plot representing log_2_ transformation of the ratio of an enriched gene’s average transcript level in the “upregulated” samples relative to that gene’s average transcript level in the “unaltered” group of samples (*x*-axis = “log ratio”), as plotted against the *p*-value (unpaired *t*-test) significance of each transcript’s enrichment in the “upregulated” samples (y-axis = log_10_
*p*-value). Light grey = *q* > 0.05; blue = *q* ≤ 0.05; (Benjamini-Hochberg correction). (**b**) *SLC6A12* gene expression levels, in tumors with upregulated vs unaltered *SLC6A13* transcript expression. Plots generated by cBioPortal, modified.

**Figure 11 ijms-21-08485-f011:**
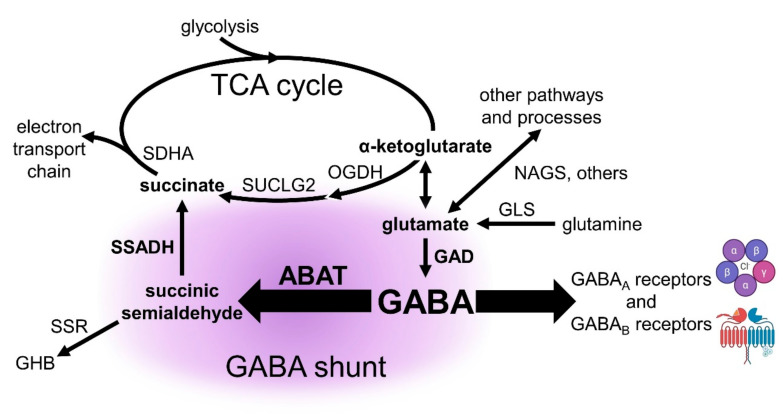
Schematic of GABA shunt, indicating consequences of upregulated *ABAT* transcripts. An increase in *ABAT* transcript, protein, and activity may result in a decrease in GABA levels, and ultimately a decrease in activation of GABA receptors. ABAT = γ-aminobutyrate aminotransferase; GAD = glutamic acid decarboxylase; GHB = gamma-hydroxybutyrate; GLS = glutaminase; NAGS = *N*-acetylglutamate synthase; OGDH = oxoglutarate dehydrogenase; SDHA = succinate dehydrogenase complex flavoprotein subunit A; SSADH = succinic semialdehyde dehydrogenase; SSR = succinic semialdehyde reductase; SUCLG2 = succinate-CoA ligase [GDP-forming] subunit beta, mitochondrial.

**Table 1 ijms-21-08485-t001:** Genes encoding GABA system proteins.

GABA Shunt Enzymes	GABA_A_ Receptor Subunits		GABA_B_ Receptor Subunits	GABA Transporters
*GAD1* (GAD67) ^1^	*GABRA1* (α1)	*GABRG1* (γ1)	*GABBR1* (B1)	*SLC6A1* (GAT-1)
*GAD2* (GAD65) ^1^	*GABRA2* (α2)	*GABRG2* (γ2)	*GABBR2* (B2)	*SLC6A11* (GAT-3)
*ABAT* (ABAT) ^2^	*GABRA3* (α3)	*GABRG3* (γ3)		*SLC6A12* (BGT-1)
*ALDH5A1* (SSADH) ^3^	*GABRA4* (α4)	*GABRD* (δ)		*SLC6A13* (GAT-2)
	*GABRA5* (α5)	*GABRE* (ε)		*SLC32A1* (VGAT)
	*GABRA6* (α6)	*GABRP* (π)		
	*GABRB1* (β1)	*GABRQ* (θ)		
	*GABRB2* (β2)	*GABRR1* (ρ1)		
	*GABRB3* (β3)	*GABRR2* (ρ2)		
		*GABRR3* (ρ3)		

Protein names given in parentheses. ^1^ GAD = glutamic acid decarboxylase. ^2^ ABAT = γ-aminobutyrate aminotransferase. ^3^ SSADH = succinic semialdehyde dehydrogenase.

**Table 2 ijms-21-08485-t002:** Relationships between *ABAT* expression levels and clinical phenotypes.

Clinical Attribute	Variables (Number of Patients)	Average *ABAT* Transcript Level (log_2_ RSEM) ± SEM	*p*-Value
Neoplasm Disease Lymph Node Stage	N0 (67)	10.8 ± 0.3	0.11
	N1 (9)	8.8 ± 0.8	
Neoplasm Disease Stage	I/II (46)	11.1 ± 0.3	0.054
	III/IV (30)	9.6 ± 0.4	
Disease-Free Status	Disease Free (34)	11.8 ± 0.3	0.014
	Recurred / Progressed (11)	9.6 ± 0.7	
Overall Survival Status	Living (51)	11.4 ± 0.3	1.0 × 10^−4^
	Deceased (27)	9.0 ± 0.5	
New Neoplasm Event Post Initial Therapy Indicator	No (34)	11.7 ± 0.3	6.0 × 10^−5^
	Yes (35)	9.2 ± 0.4	

**Table 3 ijms-21-08485-t003:** Correlations between *ABAT* methylation levels and clinical phenotypes.

Clinical Attribute	Variables (Number of Patients)	Average *ABAT* Methylation Level (β) ± SEM	*p*-Value
Overall Survival Status	Living (51)	0.66 ± 0.03	0.036
	Deceased (28)	0.78 ± 0.04	
New Neoplasm Event Post Initial Therapy Indicator	No (38)	0.60 ± 0.04	7.6 × 10^−4^
	Yes (35)	0.79 ± 0.03	

**Table 4 ijms-21-08485-t004:** Expression of transcripts encoding selected GABA shunt and shunt-proximal genes in ACC tumors with upregulated vs unaltered *ABAT.*

	*GAD1*	*NAGS*	*AKR7A2*	*SDHA*	*GLS*
Upregulated *ABAT*	8.9 ± 1.60	2.88 ± 0.72	9.60 ± 0.60	12.85 ± 0.63	11.28 ± 1.23
Unaltered *ABAT*	7.3 ± 2.2	5.67 ± 1.70	10.04 ± 0.56	12.31 ± 0.78	10.68 ± 0.86
*q*-value	3.4 × 10^−3^	1.40 × 10^−10^	0.01	0.01	0.08

Expression given as average +/− standard deviation in log_2_ RSEM. Unpaired *t*-tests, Benjamini-Hochberg correction.

**Table 5 ijms-21-08485-t005:** Tumors with upregulated *ABAT* transcripts overexpress select genes in enriched biological pathways.

Mesenchymal-to-Epithelial Transition	Interferon Gamma Signaling
*KRT8*,***GPRC5A***,***PPL***, *KCNK1*, *IRF6*, *TSPAN1*, *ARAP2*, *TTC39A*, *STAP2*, *SORL1*, *VAV3*, *SPINT2*, *ZNF165*, *SDC4*, *ADAP1*, *TPD52*, *ATP1B1*, *HNMT*, *GALNT7*	*GBP2*, *IRF6*, *NCAM1*, *HLA-A*, *HLA-B*, *IFI30*, *HLA-F*, *HLA-E*

**Table 6 ijms-21-08485-t006:** GABA system transcripts shared between ACC tumors and the NCI-H295R cell line.

**GABA Shunt Enzymes**	**GABA_A_ Receptor Subunits**		**GABA_B_ Receptor Subunits**	**GABA Transporters**
*GAD1*	*GABRA3*	*GABRG1*	*GABBR1*	*SLC6A1*
*ABAT*	*GABRA5*	*GABRD*	*GABBR2*	*SLC6A12*
*ALDH5A1*	*GABRB2*	*GABRE*		*SLC6A13*
	*GABRB3*			

For TCGA tumor data, GABA_A_ receptor subunit-encoding genes included are those determined to be relevant for expression of complete hetero- pentameric receptors in ACC tumors, while additional GABA system genes are included if median expression ≥3 log_2_ RSEM.

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
