# Peer review of "A Targeted Bioinformatics Assessment of Adrenocortical Carcinoma Reveals Prognostic Implications of GABA System Gene Expression"

_ijms, 2020, doi:10.3390/ijms21228485_

Round 1

Reviewer 1 Report

This study presents a thorough analysis of expression and methylation status of GABA (γ-aminobutyric acid) group of genes in the adrenocortical carcinoma (ACC) TCGA dataset. Authors find overexpression of some of the components of the GABA system in a subset of ACC patients (e.g. a GABA shunt gene ABAT) and show correlations of expression of some of these genes with overall survival (OS) or progression-free (PFS) survival in ACC patients (high expression of ABAT gene corresponds with OS and PFS) . Authors analyze overexpressed and underexpressed genes in ABAT overexpressing samples and also compare cortisol levels in ABAT overexpressing patients. Tumors of patients with normal serum cortisol levels have higher expression of ABAT transcripts. In addition authors investigate expression of GABAA and GABAB receptors as well as GABA transporters in ACC samples as well as their clinical prognosis. In addition, authors corroborate the expression data using publically available dataset in an ACC cell line NCI-H2595R, as well as by a validation in this cell line using qPCR..

Overall, the study shows a comprehensive analysis of the expression of GABA genes. However, authors do not provide any functional data with evidence that these receptors are key in adrenocortical carcinoma and what is their major function that could contribute to the patients’ survival effect. The analysis presented here is valid and correctly performed. It offers a strong preliminary data towards a grant or a new project. I will leave it to the editor’s decision whether this amount of data is satisfactory to be published as an article in the IJMS. Having said this, I think the bioinformatics data summarized here might be a useful source of information on GABA expression for future experimental studies and is likely to be referred to.

Some points that could improve the manuscript:

Can authors speculate what is upregulating expression of GABA genes in ACC? Is there any data available on chromatin accessibility (e.g. ATAC-seq or ChIP-seq for histone modifications) that could show accessibility of TF binding motifs in promoters/enhancers of GABA genes that could globally regulate expression of these genes?

Why ACC NCI-H295R cell line was picked? Is there any data available in other ACC cell lines?

Why data on DNMT1 upregulation and negative prognostic value for ACC patient survival is not shown? Since this data supports the rest of work – please include this in the supplementary information.

Figure 5 – please expand the figure legend and describe what is in the X axis – what are different numbers standing for – figure legend should explain what we are looking for.

When the data was accessed form cBioportal? It is important to note this detail as some data are occasionally up-dated.

Author Response

We would like to thank the reviewer for their helpful comments.  We appreciate the opportunity to clarify our work and undertake the suggested improvements. 

  • As suggested, to address the question of identifying potential common transcription factor binding motifs in the promoter regions of GABA system genes, we explored the CATA Data Base (v1.01) at http://bio.licpathway.net/cata/ (Zhou, et al., bioRxiv preprint available May 2020). This new database compiles nearly 3 million ATAC-seq peaks, searchable by gene name and cancer type.  In exploring the accessible regions within the promoters of several GABA system genes, we found that many (including GAD1, ABAT, ALDH5A1, GABRB3, GABRE, and SLC6A1) possessed binding motifs for the TF SP2, but no motifs were commonly accessible in the promoter regions of GABA system genes with co-occurring upregulation. 
  • As is now indicated in the discussion (p19) the steroidogenic NCI-H295R cell line is the most widely-used ACC cell line (>800 citations in PubMed) (Wang & Rainey, 2013). At the reviewer’s suggestion, we more thoroughly explored the data available for other ACC cell lines – such as SW-13, ACT-1, RL-251, CU-ACC1, Fang-8, and HCC1576, etc. (19 cell lines total, including H295 derivatives).  Their number of citations in PubMed range from <10 to approximately 150 (for SW-13).  Unlike for the NCI-H295R cell line, no transcript expression data is available in the NCBI’s Gene Expression Omnibus repository for these cell lines (this is now indicated in the Methods section 4.4).
  • The Kaplan-Meier survival estimation for DNMT1 is now included as Figure S2.
  • The x-axis and legend of Figure 5 have been updated to clarify the names of genes encoding GABAA receptor subunits.
  • cBioPortal data were gathered continually between January and September of 2020. This information has been added to the Methods section 4.1 alongside the cBioPortal versions accessed. 

Reviewer 2 Report

This study analyzed expressions of GABA genes, and evaluated their clinical relevance in ACC. This is an interesting topic and the authors used TCGA data to validate the existence of GABA transcripts in ACC, which is a prerequisite for those genes being potential drug targets. But I did not see strong results that supported their conclusions.

  1. Figure 1 (b). Z-score of 1 was used to group patients into “upregulated” an “unaltered”.  But it’s so strange that the violins of the upregulated and unaltered  groups still have some overlaps. 
  2. Figure 4b. Unpaired t-test followed by benjamin-Hochberge correction gave two significant q values for two plots which did not seem to be so different. Please provide more detailed statistics, including quantiles, original p-values and the number of tests used for correction. Same for figure 10(b).
  3. The discussion should be more focused instead of being too irrelevant to what’s investigated in the results. Honestly, almost every single genomics study would find dozens or hundreds of genes of clinical relevance (though only the most significant ones were reported). The number would be even larger if an arbitrary cutoff, for example top 10% versus bottom 10% or z-score >1 versus others, was used to separate samples into groups. So usually multi-sample correction was used to further remove false positives, followed by functional . Starting from a narrowed gene list, like the 30 genes GABA system, do not necessarily warrant the signals in this study were true. The authors should discuss more about technical limitations and possible future directions, instead of putting so many words on proposing or summarizing models that the authors did not even show any supportive data.
  4. The title, abstract and main text all tried to emphasize that the GABA system could be a novel target for drug development. But the whole study is all about detection of gene expression in the TCGA ACC cohort. (And, without doubt, you can always find something from pathway enrichment analyses. So further validations or data are needed if you want to make implications). The gap between detectable gene expression and being drug targets is simply too large. Functional studies of those genes in ACC or in-silico investigations of drugs-gene interactions in ACC-derived cell lines are needed if you want to deliver the information that GABA system could be drug target. Otherwise, the manuscript should be reorganized and rephrased to avoid over-interpreting their results.

Author Response

We would like to thank the reviewer for their helpful comments.  We appreciate the opportunity to clarify our work and undertake the suggested improvements.   

1.)  As violin plots are generated by kernel density estimation, their tails extend beyond the largest and smallest data points. We thank the reviewer for pointing out that this representation may be confusing to some readers, and have updated the violin plots in Figures 1b and 4a to box-and-whisker plots to avoid confusion.  Figures 1a, 5, 8, and 9 were left as violin plots as no statistical comparisons were made. 

2.)  We appreciate the reviewer’s concern that the q-values associated with Fig. 4b approach the cutoff for statistical significance. These q-values were generated by the cBioPortal user interface, wherein the Benjamini-Hochberg procedure was applied to the p-values of over 19,000 profiled genes.  A statement to this effect has been added to the Methods section 4.5.  The procedure adjusts each p-value according to its rank in an ordered arrangement of all p-values in the dataset, from smallest to largest.  These adjusted values (q) can be compared against the accepted false discovery rate, here 0.05.  Therefore, the values presented in Fig. 4b, although near the 0.05 cutoff, are statistically unlikely to be false rejections of the null hypothesis. 

As requested, to provide further quantitative information regarding this figure, the average ± standard deviation for HSD3B2 transcript expression is 8.8 ± 3.4 log2 RSEM vs 11.0 ± 4.2 log2 RSEM in tumors with upregulated vs unaltered ABAT transcripts, respectively, with an uncorrected p-value of 9.1 x 10-3.  The average ± standard deviation for NR3C1 expression is 10.4 ± 0.6 log2 RSEM vs 9.9 ± 0.9 log2 RSEM in tumors with upregulated vs unaltered ABAT transcript expression, respectively, with an uncorrected p-value of 6.0 x 10-3.  Regarding Fig. 10b, average SLC6A12 transcript expression levels ± standard deviation were 4.4 ± 1.3 and 2.9 ± 1.1 log2 RSEM in tumors with upregulated and unaltered SLC6A13, respectively (in text), and the uncorrected p-value was 2.1 x 10-6.  To avoid confusing the reader, we have not included the above additional data in the text.

3.)  The discussion has been substantially revised to remove the more speculative elements. Figures 12 and 13 and their related discussions have been removed to de-emphasize these models and avoid over-interpretation of the data.  

As the reviewer points out, attempting to identify clinically relevant genes by separating samples by an arbitrary cutoff, such as top vs bottom 10%, may return results without true clinical importance.  Surprisingly, a separation into ACC patients with the longest 10% vs shortest 10% of overall survival times revealed no genes as differentially expressed due to the low power of this small dataset. 

Many unsupervised (>100) bioinformatics studies have implicated various genes of the GABA system as relevant to cancer.  Our investigation of GABA system gene expression initially utilized the default z-score (z = 2) for the cBioPortal user interface (returning a 37% upregulation of ABAT), that was subsequently optimized.  Importantly, a variety of bioinformatics databases allow expression cutoff to be optimized for the evaluation of clinical attributes (such as the Human Protein Atlas, which by default presents patient survival data based on an optimized expression cutoff). 

A paragraph regarding the limitations inherent in bioinformatics analyses is now included in the Discussion (p20).   Additionally, we agree that the next step would be to perform functional studies to determine whether manipulating the GABA system affects cancer growth phenotypes such as proliferation, migration, and invasion.  These future experiments are described in the Discussion (p20).

4.) We recognize the interpretational limits of our bioinformatics findings, and now discuss them in the text (Discussion, p20).   Additionally, we have changed the title of the manuscript, lessened the scope of our interpretation of the data, and removed many instances of speculation.  

Round 2

Reviewer 2 Report

The authors addressed most of my points. My only concern regarding the revised manuscript is still about the q-values in Figure 4 and 10. For example, the uncorrected p-value for the comparison of HSD3B2 between the upregulated ABAT and unaltered ABAT is 0.0091. q-value was calculated by using the Benjamini-Hochberg procedure. By definition, q = p * number_of_multiple_tests/rank_of_gene. The number_of_multiple_tests here is around 19,000. Then the rank of HSD3B2 is 0.0091*19,000/0.043 = 4021. My conclusion after doing this simple calculation is that, using ABAT expression to separate ACC samples into 2 groups, more than 4,000 genes are significantly differentially expressed after multiple-test correction, which is too striking. I know that the figures and test results were generated by using cbioportal, but I am expecting to see details about how the q-values were calculated for Fig 4 and 10. Could you please repeat the tests using the original data freely available from  firehose instead of using cBioPortal? 

Author Response

Regarding Figure 4b, at the reviewer’s suggestion, we have independently downloaded the raw transcript expression data for this dataset, log-transformed the values, and utilized SPSS to perform Independent Samples T-Tests between all 20531 available genes (comparing values in samples with upregulated ABAT expression vs unaltered ABAT).  Using the results of the Levene’s Test for Equality of Variances, we selected the appropriate 2-tailed p-value for each gene.  As per the Benjamini-Hochberg procedure, we then ranked the p-values lowest-to-highest, generated a correction coefficient by dividing the number of tests (20531) by the rank of each gene, and multiplied the original p-value by this correction coefficient.  Using this procedure, 3867 genes were significantly differentially expressed between the two cohorts with q < 0.05.  For HSD3B2, p = 0.0052, rank = 3317th, m/i = 6.19, and q = 0.032.  For NR3C1, p = 0.0065, rank = 3515th, m/i = 5.84, and q = 0.038.  These values were slightly different than those generated by cBioPortal (q = 0.043 and 0.032 for HSD3B2 and NR3C1, respectively).  One possible source of this discrepancy is that cBioPortal does not assume equal variance when generating their differential expression results (as stated in their help forum on 7/6/2018).  Although our SPSS results indicate a numerically lower q-value for the differential expression of HSD3B2 between tumors with upregulated vs unaltered ABAT, we nevertheless choose to utilize the cBioPortal generated values for these figures to be consistent with the other information presented in this manuscript.

Regarding Figure 10b, we followed the same procedure described above for comparing all genes between samples with upregulated SLC6A13 transcript expression vs those with unaltered SLC6A13.  A total of 113 genes were differentially expressed between these cohorts.  For SLC6A12, p = 7.4 x 10-7, rank = 3, m/i = 6843.7, and q = 5.0 x 10-3.  Similarly, we elect to provide the cBioPortal generated values in the manuscript for the sake of consistency. 

In considering the reviewer’s comments, we agree with that finding over 4000 genes differentially expressed between these sample cohorts (as indicated on p6) is striking.  We have additionally revised the manuscript to include the number of genes utilized for downstream functional enrichment analysis, following application of a log-fold-change cutoff of ≥ |1.0|, as previously indicated in section 4.2 of the Methods.  Exact numbers of genes are now given on p7 of the Results and in section 4.2 of the Methods.